# The Role of False Self-Presentation and Social Comparison in Excessive Social Media Use

**DOI:** 10.3390/bs15050675

**Published:** 2025-05-14

**Authors:** Nor Fariza Mohd Nor, Nayab Iqbal, Azianura Hani Shaari

**Affiliations:** Centre for Research in Language and Linguistics, Faculty of Social Sciences and Humanities, Universiti Kebangsaan Malaysia (UKM), Bangi 43600, Malaysia; fariza@ukm.edu.my (N.F.M.N.); azianura@ukm.edu.my (A.H.S.)

**Keywords:** false self-presentation, social comparison, fear of negative evaluation, self-esteem, excessive social media use, gender moderation, PLS-SEM

## Abstract

Excessive social media use has become a growing concern due to its potential to affect self-perception, particularly through lowered self-esteem and a heightened fear of negative evaluation. With the increasing tendency for individuals to curate idealised online personas, understanding the psychological factors that drive this behaviour is critical. This study applies Self-Discrepancy Theory, Social Comparison Theory, and Gender Schema Theory to explain how self-perception, constant comparisons, and internalised gender norms drive online behaviours. A survey of 400 active social media users in Pakistan was conducted, and the data were analysed using Partial Least Squares Structural Equation Modelling (PLS-SEM). This study revealed that false self-presentation significantly heightened the fear of negative evaluation, which mediated its influence on excessive social media use. Social comparison, contrary to expectations, boosted self-esteem while also fuelling excessive use, as individuals sought validation online. Gender also appeared to play a moderating role, with women experiencing a stronger link between social comparison and self-esteem. However, gender did not moderate the relationship between false self-presentation and the fear of negative evaluation, suggesting similar psychological effects across genders. This study highlights that the fear of negative evaluation and social comparison-driven self-esteem are key psychological mechanisms behind excessive social media use, while gender plays a role in shaping the impact of social comparison but not false self-presentation. This study provides empirical evidence that excessive social media use is shaped by psychological factors, such as fear of negative evaluation and validation-seeking, driven by social comparison. Interventions aimed at reducing the emotional distress associated with excessive social media use should prioritise digital literacy programs that help users identify how false self-presentation and social comparison shape their fear of negative evaluation and validation-seeking behaviours, especially in appearance-driven online environments.

## 1. Introduction

Social media significantly shapes human lives by altering communication patterns, influencing self-perception, and impacting access to information. In fact, social media has become an essential part of everyday life, influencing how people interact, communicate, and view themselves ([16]; [20]; [93]; [53]). Social media platforms allow users to share carefully selected aspects of their lives, often shaping perceptions through the lens of curated content ([7]; [111]; [113]). While social media provides opportunities for connection and self-expression, concerns have emerged about its psychological impact ([8]; [74]; [85]; [107]; [112]). In particular, excessive social media use has been linked to negative emotional outcomes, such as increased anxiety and pressure to conform to unrealistic standards ([1]; [59]; [86]; [106]). Therefore, understanding the factors that contribute to this behaviour is essential for addressing its harmful effects.

One of the key areas of interest is how individuals present false versions of themselves on social media ([68]; [90]; [103]). The concept of “false self-presentation” is complex and has been addressed by various scholars, particularly within psychology and sociology. Donald W. Winnicott, a British psychoanalyst, introduced the concept of the “false self” in 1960. Winnicott described the false self as a defensive facade, where individuals present a persona that complies with external expectations, concealing their true feelings and desires ([26]). Similarly, [37] ([37]) conceptualised self-knowledge as a “self-digest” that helps individuals regulate their behaviour by summarizing their relationships with the world and the personal consequences of these interactions. This self-regulation can shape how individuals engage in self-presentation, particularly on social media, where they selectively curate aspects of their identity.

[104] ([104]) identified two distinct forms of false self-presentation on social media: lying behaviours and liking behaviours. Lying behaviours involve sharing untruthful status updates or creating deceptive profiles to present a fabricated version of oneself. On the other hand, liking behaviours refer to engaging with content—such as liking posts—in a way that does not genuinely reflect one’s true feelings or opinions. Both behaviours are often driven by a desire for social approval or popularity, leading individuals to curate an online persona that may not align with their authentic selves.

False self-presentation, therefore, reflects how some social media users carefully craft their online personas to project an idealised or inauthentic version of their lives ([64]). This drive to manage one’s online image creates pressure to conform to expected standards, leading to a heightened fear of negative evaluation ([2], [3]; [10]; [91]). This fear can drive users to engage more frequently and intensely with social media as they strive to maintain or enhance their false self-presentation ([27]; [43]; [71]).

Visual messages on the Grindr mobile app’s default pictures, as reported in a study by [28] ([28]), revealed users self-branding themselves to personify whom they encompass as a human in order to attract whomever their intended audience is, which suggests how individuals become overly concerned about how others perceive them, fearing judgment or disapproval if their curated persona falls short of these expectations ([49]). Therefore, individuals posted photos that make them appear more presentable and attractive.

A study by [72] ([72]) also affirms that social networking sites allow users to have greater control over their self-presentation, as users create personal profiles and control what images and/or posts they share to represent themselves. While this study does not directly analyse selfies, the prominence of selfie-sharing is a well-documented feature of self-presentation on social media. As noted by [72] ([72]), users often curate personal images—including selfies—to construct a desirable online identity. This reflects the broader trend of visual self-presentation, which plays a significant role in how individuals manage their perceived image in digital spaces.

Social comparison is another significant factor contributing to excessive social media use. Humans have a natural tendency to compare themselves with others, and social media platforms amplify this behaviour ([42]; [61]; [77]). Users are constantly exposed to the carefully curated content of others, which often presents an unrealistic or overly positive image of life ([4]; [5]; [80]). When people compare themselves to these idealised versions of others, it can lead to feelings of inadequacy, particularly if they believe they do not measure up ([45]; [99]; [101]; [110]). This process can negatively affect self-esteem, as individuals internalise the perceived gap between their own lives and the lives they see online. Furthermore, in this study, excessive social media use refers not only to frequent use, but to engagement patterns characterised by difficulty disconnecting, compulsive checking, and reliance on social media for self-worth or emotional regulation—behaviours that go beyond casual or routine use.

This study also considers the role of gender as a moderating factor. Gender differences have been widely observed in the ways individuals engage with social media, particularly in relation to self-presentation and social comparison. Research suggests that women, more than men, may experience greater pressure to present themselves in a certain way online, leading to heightened concerns about appearance and social acceptance ([11]; [35]; [55]; [57]).

This study primarily aims to examine the relationships between self-presentation, social comparison, and excessive social media use, with a specific focus on the moderating role of gender. The objectives of this research are to determine whether self-presentation influences excessive social media use through its association with fear of negative evaluation and to assess the impact of social comparison on excessive social media use via its relationship with self-esteem. Additionally, this study seeks to evaluate whether gender moderates the relationship between self-presentation and the fear of negative evaluation, as well as the relationship between social comparison and self-esteem.

Although previous research has examined the role of self-presentation and social comparison in online behaviour, limited attention has been given to the false nature of self-presentation and its link to the fear of negative evaluation as a pathway to excessive social media use. Similarly, while social comparison has been studied in relation to self-esteem, its indirect influence on excessive use through self-esteem remains underexplored. Furthermore, most existing studies treat gender as a demographic variable, rather than investigating its moderating role. This study addresses these gaps by proposing a moderated mediation model that integrates Self-Discrepancy Theory, Social Comparison Theory, and Gender Schema Theory to explain how psychological processes and gender differences contribute to excessive social media engagement.

This research fills a gap in the literature by examining the relationships between false self-presentation, social comparison, and excessive social media use, while considering how gender may influence these patterns. Social media plays a significant role in shaping how people see themselves and interact with others. This study offers insight into the psychological factors—such as the fear of negative evaluation and validation-seeking—that drive high engagement. Recognising how gender influences these behaviours can help in creating more focused strategies to reduce the emotional strain linked to social media use.

## 2. Literature Review

### 2.1. Self-Discrepancy Theory

Self-Discrepancy Theory posits that individuals hold different self-concepts, and the discrepancies between these self-concepts can lead to distinct emotional and psychological outcomes ([36]). The three central components of this theory are the actual self (the attributes individuals believe they currently possess), the ideal self (the attributes individuals aspire to have, representing hopes and desires), and the ought self (the attributes individuals believe they should have, based on obligations and responsibilities).

When individuals perceive a significant gap between their actual self and either their ideal or ought self, they experience discomfort and emotional distress ([39]; [82]). The theory posits that a discrepancy between the actual self and the ideal self often leads to emotions such as disappointment, dissatisfaction, or sadness, as the individual feels they are not achieving their personal aspirations. On the other hand, a discrepancy between the actual self and the ought self typically leads to anxiety, guilt, or fear, as individuals feel they are not meeting societal or external expectations ([48]).

The emotional responses resulting from these discrepancies are theorised to motivate individuals to reduce the gap between their different self-concepts ([18]; [94]; [98]). The greater the discrepancy, the more intense the emotional discomfort ([38]). Higgin’s Self-Discrepancy Theory increases understanding of psychological response to self-discrepancies, whereby individuals differ in their sensitivity to these discrepancies, which can influence how they respond. Those who are more focused on their ideal self are likely to experience heightened negative emotions related to aspiration failures, while those more focused on their ought self can trigger the experience of increased anxiety and guilt when they perceive societal shortcomings.

The principles of Self-Discrepancy Theory are particularly relevant in the context of social media use, where the distinctions between the actual self, ideal self, and ought self-become increasingly pronounced ([5]; [52]; [73]). Social media platforms provide users with opportunities to present curated and often idealised versions of themselves because the public nature of these platforms allows individuals to showcase only the aspects of their lives that align with their ideal self—their aspirations, achievements, and highlights ([12]; [69]; [108]).

In addition to comparing themselves with others online, users may also experience internal conflict when comparing their own false or idealised online persona with their real, offline self. This internal comparison can heighten self-awareness and lead to discomfort or self-criticism, particularly when the gap between one’s actual and projected identities becomes more evident. In this case, social media may not only expose users to idealised images of others but also confront them with an unrealistic version of themselves that they feel compelled to maintain.

The curation of self-presentation on social media—through the selective sharing of achievements, appearances, or aspirational moments—can exacerbate the perceived gap between the actual self and the ideal self. This widened discrepancy often intensifies emotional strain, leading to dissatisfaction or frustration when individuals compare their everyday realities to the polished versions they showcase online.

Furthermore, social media amplifies the pressures related to the ought self, where users feel the need to conform to societal norms, peer expectations, and cultural standards ([4]; [17]; [102]). The presence of likes, comments, and shares as forms of public validation intensifies the drive to meet these external expectations ([40]; [46]; [56]). When individuals perceive that their self-presentation on social media does not align with what they believe is expected of them, it can lead to anxiety, guilt, or fear of negative evaluation ([14]; [63]; [78]). Thus, the discrepancies between the actual, ideal, and ought selves are not only heightened by the constant comparisons inherent to social media but also amplify the compulsion to engage more frequently with these platforms in an attempt to bridge the gap between these self-concepts.

### 2.2. Self-Presentation

Self-presentation refers to the process by which individuals attempt to control the impressions they make on others ([51]). This concept, rooted in Impression Management Theory ([32]), involves consciously curating behaviours, images, and information to influence how one is perceived. On social media, this act of self-presentation often takes the form of highly selective content sharing—posting photos, statuses, or achievements that portray an idealised version of one’s life ([76]; [83]; [114]). However, this idealised self-presentation may not accurately reflect the individual’s true circumstances, leading to what is commonly referred to as false self-presentation ([64]; [66]; [81]).

Although this study focuses on false self-presentation, it is important to note that not all users portray themselves positively on social media. Some individuals may intentionally present themselves in a negative light, often to seek empathy, express vulnerability, or attract social support. While this form of presentation falls outside the scope of the current model, it reflects the broader range of identity strategies used online.

Self-presentation behaviour can be measured via language, as revealed in a study by [33] ([33]). They found that the frequent use of positive emotions and words, as well as words related to achievement, in status messages are ingratiation and competence strategies ([32]) that are frequently used by the participants in their attempt to present themselves as competent and likeable.

False self-presentation occurs when individuals misrepresent themselves or exaggerate aspects of their lives to align more closely with perceived societal standards or expectations ([41]; [70]; [97]). This can include posting heavily edited or staged photos, presenting one’s life as more exciting or glamorous than it is, or sharing content that does not reflect one’s true values or emotions. The pervasive nature of social media has increased opportunities for individuals to engage in false self-presentation, as platforms like Instagram or Facebook offer spaces for users to construct a narrative of themselves that may be far removed from their actual reality ([47]; [60]; [100]).

The practice of false self-presentation is closely linked to the need for validation and approval from others. Individuals may engage in this behaviour to gain social acceptance, approval, or admiration, often relying on likes, comments, and shares to measure the success of their self-presentation efforts ([22]; [24]). Research by [66] ([66]) indicates that people who frequently engage in false self-presentation tend to use social media more excessively, as they seek continuous feedback to maintain their curated online persona. This constant need to manage and monitor one’s online image can lead to excessive social media use, where users spend significant time ensuring their content aligns with their idealised self-presentation and checking for feedback.

In addition to its impact on social media usage, false self-presentation also intensifies the fear of negative evaluation. The more effort individuals put into curating an idealised or false version of themselves, the more anxious they become about being judged or criticised for failing to meet those expectations ([15]; [31]; [88]). [2] ([2]) found that people who experience high levels of fear of negative evaluation are more likely to use social media to control how others view them, further entrenching the cycle of self-presentation and social media dependence. Therefore, it can be hypothesised that as individuals engage in false self-presentation, they become increasingly concerned with how others perceive them, driving up anxiety and fear that they will be exposed as inauthentic or judged for not living up to the image they project.

While false self-presentation often involves exaggeration or misrepresentation, idealised self-presentation is not always entirely false. Idealised self-presentation involves selectively highlighting positive traits, while false self-presentation refers to intentionally portraying traits or behaviours that do not reflect one’s actual self. Although these two forms of self-presentation can overlap, the key difference lies in intent and authenticity. As individuals adopt false self-presentations, concerns about being exposed or judged for inauthenticity tend to grow, further fuelling anxiety and the fear of negative evaluation.

This fear of negative evaluation also leads to excessive social media use. When individuals fear being judged or criticised, they often engage in behaviours that reduce this anxiety, such as compulsively checking for feedback, deleting posts that don not receive enough validation, or continuously adjusting their self-presentation to align with perceived social expectations ([75]; [92]). Therefore, it can be hypothesised that

**H1:** *False self-presentation is positively associated with excessive social media use*.

**H2:** *False self-presentation is positively associated with the fear of negative evaluation*.

**H3:** *The fear of negative evaluation is positively associated with excessive social media use*.

### 2.3. Social Comparison

Social comparison refers to the process by which individuals evaluate themselves in relation to others. Social Comparison Theory suggests that people have an inherent drive to compare themselves with others to gauge their own abilities, success, and worth ([29]). On social media platforms, this comparison is constant, as users are regularly exposed to idealised representations of others’ lives. These curated posts often showcase the most positive aspects of someone’s life—whether it’s physical appearance, achievements, or social connections—resulting in what is known as upward social comparison, where individuals compare themselves to those they perceive as superior in some way ([96]).

Upward social comparison can have a significant impact on users’ emotional and psychological well-being, often leading to feelings of inadequacy or dissatisfaction with one’s own life ([54]; [77]). The frequency of these comparisons on social media platforms contributes to excessive social media use, as users feel compelled to keep up with the seemingly perfect lives of others ([58]; [61]). This constant comparison, particularly when upward in nature, often exacerbates feelings of anxiety or dissatisfaction, further encouraging the excessive use of social platforms ([23]; [76]; [96]).

The relationship between self-esteem and excessive social media use is equally complex. Individuals with low self-esteem may use social media more frequently in an attempt to seek validation and improve their self-worth through likes, comments, and social engagement ([19]; [67]; [89]). Conversely, those with high self-esteem may engage with social media in a more balanced way, using it to reinforce their positive self-image but without becoming dependent on the feedback of others ([62]; [84]; [95]). However, even individuals with high self-esteem can be drawn into excessive use if they rely on social media for continuous validation or if they engage in upward social comparisons that challenge their self-image. Based on these patterns, the following hypotheses were developed to examine the role of social comparison and self-esteem in excessive social media use.

**H4:** 
*Social comparison is positively associated with excessive social media use.*


**H5:** 
*Social comparison is negatively associated with self-esteem.*


**H6:** *Self-esteem is negatively associated with excessive social media use*.

### 2.4. Gender Schema Theory

Gender Schema Theory suggests that individuals organise information and experiences based on culturally defined gender norms ([6]). These internalised schemas guide how people present themselves and respond to social feedback. Men and women internalise different social expectations based on their gender ([109]), which affects how they engage in self-presentation and respond to social comparison on platforms like social media.

According to Gender Schema Theory, women are often socialised to prioritise appearance and social approval, leading them to engage more frequently in self-presentation that aligns with societal standards of beauty and success ([87]). As women internalise these gender-based expectations, they may experience a heightened vulnerability of negative evaluation when their self-presentation does not align with these standards. Men, however, whose gender schemas prioritise achievement and independence, exhibit less sensitivity to the social pressures inherent in self-presentation.

Gender Schema Theory also posits a psychological vulnerability, whereby women, who are driven by appearance-related norms, tend to engage in social comparisons—stemming from the emphasis placed on physical appearance in the female gender schema.When women experience this vulnerability, particularly in upward social comparisons, it may lead to devastating impacts on self-esteem. Psychologically, for men, whose self-worth may be less rooted in appearance and more in markers of competence and success, their self-esteem remains less vulnerable to the fluctuations of social comparison, creating a weaker psychological link.

When women compare themselves to others, particularly in upward social comparisons, they are more likely to experience declines in self-esteem, as their gender schema places greater importance on conforming to societal ideals ([34]). For men, whose self-worth may be tied less to appearance and more to other factors like competence or success, the relationship between social comparison and self-esteem is likely to be weaker.

While Gender Schema Theory provides a valuable lens for understanding how internalised gender norms influence self-presentation and social comparison, it is important to acknowledge that such norms are socially constructed and may not apply universally. The theory operates within a binary framework that assumes distinct psychological orientations for men and women, yet gender identity and behaviour are far more fluid and context-dependent. Individuals may adopt, resist, or redefine these roles in diverse ways, and patterns of self-presentation or comparison are not fixed or exclusive to any one gender. The present study uses Gender Schema Theory to explore commonly observed trends but does not claim that these reflect inherent differences between men and women, nor does it disregard the roles and experiences of non-binary or gender-diverse individuals.

Based on these theoretical assumptions, the following hypotheses were proposed to test the moderating role of gender within the model:

**H7:** 
*Gender moderates the relationship between false self-presentation and the fear of negative evaluation, such that the relationship will be stronger for women than for men.*


**H8:** 
*Gender moderates the relationship between social comparison and self-esteem, such that the relationship will be stronger for women than for men.*


Figure 1 shows how the Self-Discrepancy Theory, Self-Presentation Theory, and Gender Schema Theory are related to each other in the present study.

## 3. Method

This study was conducted using a quantitative methodology deploying a survey method. This approach allowed for the collection of structured data that could be statistically analysed to identify patterns and relationships between the constructs under investigation. The sampling process involved recruiting 400 participants from Pakistan, selected through purposive sampling. A total of 432 responses were initially collected. After removing incomplete submissions and screening for inconsistencies or a response bias, 400 valid responses were retained for the analysis. No financial incentives were offered; however, participants were informed about the academic purpose of the study and assured of the confidentiality of their responses.

Purposive sampling is a non-probability sampling technique in which researchers deliberately select participants based on specific characteristics that align with the research objectives ([9]). This method is commonly used in studies where the target population possesses particular traits relevant to the research question ([13]). To fulfil the requirement of the study, the participants were required to be active social media users, defined as individuals who accessed social platforms at least once daily, ensuring that only those with regular engagement were included in the study. This criterion was critical for capturing authentic, ongoing patterns of social comparison and self-presentation.

The data collection process involved administering an online survey to participants, which was distributed via social media platforms, such as Facebook, Instagram, and WhatsApp groups, targeting active users of social media, specifically Facebook, in Pakistan. The selection criteria have been mentioned earlier. The participants were provided with a clear consent form explaining the purpose of the study and assuring confidentiality. After consent was obtained, the participants were directed to complete the survey, which took approximately 15–20 min.

For the data analysis, Partial Least Squares Structural Equation Modelling (PLS-SEM) was employed to test the study’s hypotheses and assess the relationships. PLS-SEM was chosen for its ability to handle complex models with multiple constructs and its robustness in working with smaller sample sizes. The analysis included evaluating the measurement model for reliability and validity, followed by testing the structural model to assess path coefficients, R-squared values, and the significance of the hypothesised relationships.

### 3.1. Instruments

This study employed a variety of well-established measures to assess the core constructs. All the measures used in the study were adapted to test the hypotheses, and each construct was assessed using a 7-point Likert scale. Compared to a 5-point scale, a 7-point scale provides greater variability and precision in capturing the intensity of responses while still being easy for participants to understand ([21]). Research suggests that 7-point scales improve reliability and validity by allowing respondents to express more varied opinions without overwhelming them with too many options ([44]).

False self-presentation was measured using the Self-Presentation on Facebook Questionnaire (SPFBQ), which is a 17-item scale developed by [65] ([65]), including real, ideal, and false selves. Example items included “What I put on Facebook was a fairly comprehensive representation of myself” and “My statements about my feelings on Facebook were always honest”.

Social comparison was measured using 11 items of the Iowa–Netherlands Social Comparison Orientation Scale ([30]). The sample items included “I always pay a lot of attention to how I do things compared with how others do things” and “I always like to know what others in a similar situation would do”.

The fear of negative evaluation was assessed using the Brief Fear of Negative Evaluation Scale (BFNES) developed by [50] ([50]), which is an 11-item self-report measure that evaluates individuals’ discomfort and fear of being judged negatively by others. For example, “I am usually worried about what kind of impression I make” and “I often worry that I will say or do the wrong things”.

Self-esteem was measured using the widely used Rosenberg Self-Esteem Scale (RSE), a 10-item scale where participants rated their overall self-esteem, as in [79] ([79]). Two example items were “I feel that I have a number of good qualities” and “I take a positive attitude toward myself”.

Excessive social media use, specifically Instagram, was assessed using an adapted version of the nine-item Social Media Use Questionnaire (SMUQ) developed by [105] ([105]). The scale captures compulsive engagement, salience, withdrawal, and conflict associated with social media behaviours. Sample items included “I often check social media without intending to and lose track of time” and “I feel restless or anxious when I haven’t checked social media for a while”.

### 3.2. Demographic Information

The sample consisted of 400 participants, with 55.25% female participants (n = 221) and 44.75% male (n = 179). The majority of the participants (45.75%) were between 18–25 years, followed by 26–35 years (34.50%), while those aged 36–45 years made up 14.25% of the sample, and the participants aged 46–55 and above 55 each accounted for 2.75%. The mean age of the participants was 28.4 years (SD = 8.3), indicating a sample primarily composed of young adults—consistent with prior research identifying this group as the most active demographic on social media platforms.

Regarding education, 43.25% held a bachelor’s degree, 20.75% had an intermediate level education, 19.75% had a master’s degree, and a small portion (5.25%) held a PhD, with 4.25% reporting no formal education. In terms of income, 34.25% reported a monthly household income between PKR 50,000–PKR 250,000, followed by 27.25% earning PKR 250,000–PKR 500,000. Income was included as a demographic variable to allow for the consideration of socioeconomic diversity within the sample. Although not the primary focus of this study, reporting income offers contextual insight and may support future research exploring how socioeconomic status influences patterns of self-presentation or validation-seeking behaviour on social media.

Social media usage revealed that 40.50% of the participants spent 1–3 h daily on social media, while 29.25% reported spending 3–5 h, and 15.75% used social media for less than an hour per day. The results are displayed in Table 1.

## 4. Findings

### 4.1. Reliability and Validity

The reliability and validity of the constructs were assessed using Cronbach’s alpha, composite reliability (CR), and the average variance extracted (AVE). All the constructs demonstrated strong internal consistency, see Table 1, with the Cronbach’s alpha values ranging from 0.74 to 0.936, indicating that the measures are reliable. The composite reliability values were also above the acceptable threshold of 0.7, further confirming the reliability of the scales. The AVE values, which measure the amount of variance captured by the construct relative to the variance due to measurement error, ranged from 0.563 to 0.796. Meanwhile, the AVE for all constructs met or exceeded the recommended threshold of 0.50, indicating adequate convergent validity. The results are presented in Table 2.

Table 3 shows the discriminant validity between the key constructs using the Heterotrait–Monotrait Ratio of Correlations (HTMT) ratio. Discriminant validity ensures that each construct in the model is distinct from others, meaning they measure different aspects of social media behaviour and psychological outcomes. All the values, as shown in the table, between the constructs were below the threshold of 0.90, confirming that the constructs are empirically distinct. This indicates that there is no significant overlap between the constructs, validating that each is measuring a unique concept in the model.

### 4.2. Structural Model Evaluation

The path model (see Figure 2) illustrates the relationships between false self-presentation, social comparison, the fear of negative evaluation, self-esteem, and excessive social media use. The results indicate that individuals who frequently engage in false self-presentation are significantly more likely to experience a heightened fear of negative evaluation, with a path coefficient of β = 0.380 (*p* = 0.000) (H1). Furthermore, the fear of negative evaluation significantly predicts excessive social media use (β = 0.213, *p* = 0.000) (H2). However, the direct relationship between false self-presentation and excessive social media use (β = 0.043, *p* = 0.300) (H3) is non-significant, indicating that engaging in false self-presentation does not necessarily translate to excessive social media use.

Similarly, the results highlight the role of social comparison in shaping self-esteem, with a path coefficient of β = 0.551 (*p* = 0.000) (H4). This indicates a strong and statistically significant relationship, suggesting that individuals who frequently compare themselves to others tend to have higher self-esteem. Furthermore, social comparison significantly predicts excessive social media use (β = 0.233, *p* = 0.000) (H5), indicating that individuals who engage in frequent social comparison are more likely to use social media excessively. However, the direct relationship between self-esteem and excessive social media use (β = −0.002, *p* = 0.969) (H6) is non-significant, meaning that an individual’s level of self-esteem does not meaningfully influence their likelihood of excessive social media use.

Table 4 shows the R-square and R-square-adjusted values for the fear of negative evaluation, self-esteem, and excessive social media use, which represent the amount of variance in these constructs explained by the model. The R-square value for the fear of negative evaluation is 0.145, meaning that 14.5% of the variance in the fear of negative evaluation is explained by the predictors in the model. Self-esteem has a higher R-square value of 0.304, indicating that 30.4% of the variance in self-esteem is accounted for by the model. Furthermore, the R-square for excessive social media use is 0.154, meaning that the model explains 15.4% of the variance in this construct.

### 4.3. Moderating Role of Gender

The moderating effect of gender was assessed for two key relationships in the model (see Table 5): the impact of false self-presentation on the fear of negative evaluation (H7) and the effect of social comparison on self-esteem (H8). A multi-group analysis (MGA) was performed to explore whether gender significantly influenced these relationships.

The results show that gender did not significantly moderate the relationship between false self-presentation and the fear of negative evaluation. This means that whether someone is male or female did not change how strongly false self-presentation led to the fear of negative evaluation. While the path coefficients, which indicate the strength of the relationship, were slightly different for males (β = 0.390) and females (β = 0.350), this difference was not statistically significant. Both males and females, therefore, experienced a similar increase in the fear of negative evaluation when engaging in false self-presentation. Therefore, our hypothesis (H7) that gender would show a stronger link between these two factors for females was not supported.

On the other hand, the findings indicate that gender significantly influenced how social comparison affected self-esteem. Specifically, the relationship between social comparison and self-esteem was much stronger for females than for males. This is shown by the path coefficients: β = 0.600 for females and β = 0.410 for males. The fact that this difference was statistically significant means that females’ self-esteem is more heavily influenced by comparing themselves to others than males’ self-esteem. This supports our hypothesis (H8). Furthermore, we observed that males were more likely to engage in false self-presentation, while females were more likely to engage in social comparison. These findings align with common gendered patterns in social behaviour.

## 5. Discussion

The results provide valuable insights into the psychological drivers behind excessive social media engagement, highlighting the various aspects of self-presentation, comparison with others, and emotional responses related to self-worth. One of the key findings of this study is the significant relationship between false self-presentation and the fear of negative evaluation.

An unexpected finding is that while false self-presentation significantly influences the fear of negative evaluation, it does not have a direct effect on excessive social media use (H1), as the path between these variables was found to be non-significant. The results show that individuals who engage in false self-presentation are more likely to experience the fear of negative evaluation, supporting earlier findings by [15] ([15]) and [31] ([31]), who observed similar patterns of anxiety linked to inauthentic online portrayals.

Consistent with the H2 hypothesis, the curated online image becomes an “ideal self” in which a carefully constructed self-presentation is developed to meet perceived societal expectations. When individuals engage in false self-presentation, they create a gap between their “actual self” and this idealised online persona or image. This discrepancy triggers negative emotions, particularly anxiety and a fear of criticism, as previously observed in studies by [15] ([15]), [31] ([31]) and [88] ([88]). Our findings align with these earlier results, suggesting that self-discrepancy continues to be a key psychological driver in social media behaviour. The constant pressure to maintain this fabricated image creates anxiety, as individuals fear their “actual self” will be seen as inadequate or unacceptable. This fear of negative evaluation, therefore, is not merely a social anxiety, but a direct consequence of the psychological discomfort generated by the self-discrepancy inherent in false self-presentation. This aligns directly with Higgins’ Self-Discrepancy Theory (1987), which posits that individuals experience emotional distress when there are perceived discrepancies between their actual self and their ideal or ought selves.

Rather than having a direct link between false self-presentation and increased social media use, our findings suggest that the fear of negative evaluation mediates the relationship between false self-presentation and social media use, indicating that users turn to platforms and engage in self-presentation to manage this fear and seek validation (H3). This finding could be explained by the idea that people use social media not only as a tool for presenting an idealised self, but also as a space to curate their online image to gain reassurance and seek approval from their peers. This suggests a drive to manage fear by increasing their engagement with social media.

Furthermore, the results show that social comparison significantly influences excessive social media use (H4). In essence, individuals who frequently compare themselves to others on social media platforms are more likely to spend excessive amounts of time on social media platforms. This is likely because constantly viewing others’ curated online lives can trigger feelings of inadequacy in order to improve their self-presentation, or the desire for validation. People may feel compelled to stay engaged to enhance their own online image, monitor how they measure against others, or seek the approval of their peers. This aligns with prior work by [61] ([61]), who found that frequent exposure to idealised online content undermines users’ self-perception, particularly when appearance-based comparisons are involved. This behaviour reflects a common trend: the pressure to “keep up” with perceived online perfection often leads to increased social media usage.

A particularly unexpected result was the positive correlation between social comparison and self-esteem (H5). This is contrary to the commonly observed pattern in prior studies, where upward social comparison has been associated with feelings of inadequacy and decreased self-esteem ([77]; [99]). It is thus possible that in certain contexts, people engage in upward social comparison and feel inspired or motivated. They might see themselves as performing well in comparison, or they might use the success of others as a benchmark to strive towards, thus boosting their self-esteem.

This finding somewhat questions the conventional view of social comparison as inherently detrimental. It suggests that the effect of social comparison is highly context-dependent, contingent upon individual interpretation and psychological mechanisms. For certain individuals, engaging in social comparison can serve as a source of inspiration and validation, leading to an enhancement of self-esteem. This may occur because social media platforms offer a curated reflection of success and positive attributes that can potentially reinforce these positive comparative experiences, contributing to an observed increase in self-esteem with heightened comparison.

Moreover, the relationship between self-esteem and excessive social media use (H6) was found to be insignificant in this study. This suggests that contrary to what might be expected, self-esteem does not directly influence how much individuals engage with social media. In other words, whether individuals have high or low self-esteem does not significantly affect their likelihood of using social media excessively. This result challenges the assumption that individuals with lower self-esteem would necessarily turn to social media as a means of seeking validation or approval.

Previous research has distinguished between individuals with high self-esteem, who tend to use social media in a more balanced and self-assured manner, and those with low self-esteem, who are more likely to engage in validation-seeking behaviours ([19]; [25]; [94]). The non-significant relationship observed in our study may reflect this distinction, suggesting that these underlying motivational differences do not necessarily translate into excessive usage patterns.

One possible explanation for this finding could be that excessive social media use is driven by other factors, such as the fear of negative evaluation or the desire to engage in social comparison, rather than self-esteem alone. It could also be that individuals with different levels of self-esteem use social media for various purposes, not all of which lead to excessive use. For example, individuals with higher self-esteem might use social media in a balanced way for networking or self-expression, while those with lower self-esteem may engage for validation, but neither group necessarily uses it excessively. This points to the complexity of social media behaviour, suggesting that self-esteem alone may not be a strong predictor of excessive use.

One of the most significant contributions of this study is its examination of gender as a moderating factor in the relationships between false self-presentation, social comparison, and the psychological outcomes of fear of negative evaluation and self-esteem. Interestingly, the moderating effect of gender was not significant in the relationship between false self-presentation and the fear of negative evaluation (H7). Although women may engage more in curated, appearance-based content, the fear of being judged for inauthentic self-presentation appears to be shared across the genders. This suggests that the psychological burden of maintaining an idealised online identity—especially when it departs from one’s actual self—is not gender-exclusive. While the motivations and forms of self-presentation may differ between men and women, the internal pressure to uphold a projected identity may lead to similar levels of fear of negative evaluation. These findings highlight how social media environments create shared anxieties, even when gendered expectations diverge.

The results show that gender plays a defining role, particularly in the relationship between social comparison and self-esteem (H8), which was stronger for women than for men. This indicates that women may be more vulnerable to the effects of upward social comparison, especially when it comes to appearance and lifestyle portrayals. The stronger link between social comparison and self-esteem for women is also reflected in studies by [34] ([34]) and [64] ([64]), which show that appearance-focused comparisons tend to affect women’s self-worth more significantly than men’s.

This finding is consistent with Gender Schema Theory, which suggests that women are socialised to seek social approval and place greater value on appearance. As a result, they may engage more frequently in appearance-based comparisons and experience sharper declines in self-esteem when they perceive themselves as falling short of the often-idealised images portrayed on social media. The emphasis placed on visual content across most platforms may further amplify this effect. A heightened sensitivity to likes, comments, and visual comparison may explain why women in this study reported lower self-esteem after engaging in these comparisons, even if the platform engagement is similar across genders.

This study makes two important contributions. First, it highlights the role of false self-presentation in increasing the fear of negative evaluation, which in turn drives excessive social media use. This supports the idea that psychological discomfort, rather than just habitual use, underlies high engagement on social media platforms. Second, the findings show that gender moderates the relationship between social comparison and self-esteem, with women more negatively affected by upward comparisons. This suggests that gendered expectations around appearance and approval shape how users experience and respond to social content.

The findings of this study have several key theoretical and practical implications. The significant relationship between false self-presentation and the fear of negative evaluation supports established theories of self-presentation, such as Goffman’s Impression Management Theory, by demonstrating how digital environments intensify the pressure to curate an idealised self. This suggests that social media is not just a platform for communication but also a space where individuals engage in performative behaviour that can have real psychological consequences.

This study provides further support for Self-Discrepancy Theory, demonstrating how the social-media-amplified gap between one’s actual self and ideal self can drive negative emotional outcomes, such as the heightened fear of judgment. This emphasises the significant psychological effect of digital identity construction. The findings highlight how digital identity construction through false self-presentation may contribute to excessive social media use. These results also call for further research to explore how sustained engagement in such behaviours relates to psychological outcomes connected to self-worth and social approval.

## 6. Implications

The findings of this study show that excessive social media use is shaped by psychological mechanisms, such as the fear of negative evaluation and the pursuit of validation through social comparison. While false self-presentation alone does not lead directly to excessive use, it indirectly contributes by increasing users’ anxiety about being judged. Interestingly, social comparison was associated with both higher self-esteem and increased social media use, suggesting that users may feel empowered by comparison but still remain dependent on online validation. These results highlight the need for awareness-based interventions that help users recognise how their self-presentation strategies and comparison habits affect their emotional engagement with social media. Digital literacy programs, in this context, should not only address unrealistic portrayals online but also encourage users, particularly women, to critically reflect on how these interactions shape their self-esteem and behaviour.

In addition, social media platforms could consider improving the transparency around engagement metrics and algorithmic content delivery. When users better understand how content is selected and amplified, they may be less likely to internalise unrealistic comparisons or rely on external validation. These platform-level changes, combined with targeted education, may help reduce compulsive use and its underlying psychological drivers.

## 7. Limitations and Future Directions

This study has several limitations. The cross-sectional design restricts the ability to establish causal relationships between false self-presentation, social comparison, and excessive social media use. Therefore, longitudinal research would provide a better insight into how these dynamics unfold over time. Another limitation involves the reliance on self-reported data, which may introduce a bias, as the participants could misreport their social media behaviour. Using objective measures, like tracking actual social media activity, would enhance the accuracy of the findings.

This study’s sample was drawn from a specific geographic and cultural context, which may limit the generalizability of the results. Social media behaviour and the pressures associated with self-presentation can differ significantly across cultures. Future research should explore how these relationships vary in different cultural settings to broaden the applicability of the findings.

In terms of future directions, it is important to investigate the long-term psychological effects of false self-presentation, particularly regarding its impact on mental health. Additionally, future research can consider assessing the effectiveness of interventions, such as digital literacy programs and changes to social media platform features, to alleviate the negative psychological effects highlighted in this study.

A more significant understanding of social media behaviours and their outcomes necessitates the in depth exploration of demographic factors, particularly age, gender, and socioeconomic status. For instance, age-related differences in cognitive development and life experiences may influence how individuals perceive and engage with online content. Younger users might be more susceptible to social comparison pressures, while older adults may prioritise information seeking. Similarly, gender roles and societal expectations can shape online self-presentation and interaction patterns. Moreover, while Gender Schema Theory helped explain the observed gender differences, its binary framework limits how these findings can be applied beyond male–female categories. Future studies should consider alternative approaches that account for gender diversity and variations in gender identity.

## Figures and Tables

**Figure 1 behavsci-15-00675-f001:**
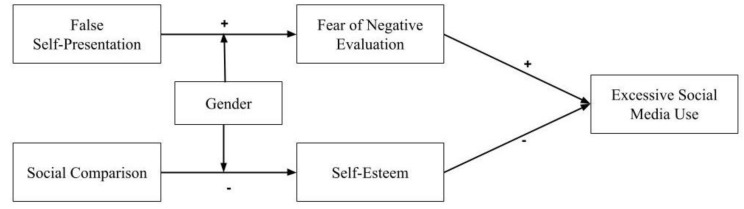
Conceptual framework.

**Figure 2 behavsci-15-00675-f002:**
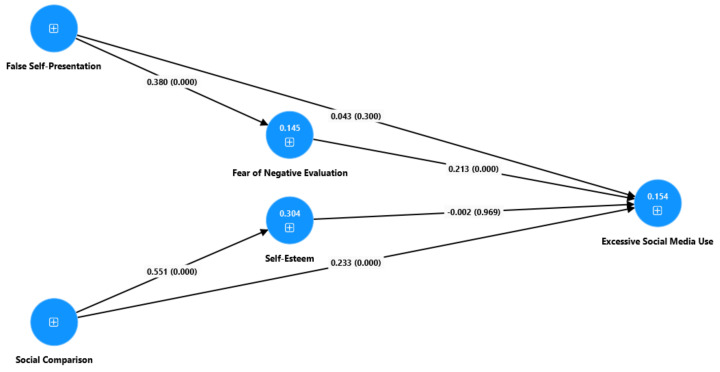
Structural equation model.

**Table 1 behavsci-15-00675-t001:** Demographic profiles.

Demographic Variable	Category	N	Percentage (%)
Gender	Male	179	44.75%
	Female	221	55.25%
Age	18–25 years	183	45.75%
	26–35 years	138	34.50%
	36–45 years	57	14.25%
	46–55 years	11	2.75%
	55 and above	11	2.75%
Education Level	No formal education	17	4.25%
	Diploma	47	11.75%
	Intermediate	83	20.75%
	Bachelor’s	173	43.25%
	Master’s	79	19.75%
	PhD/Doctorate	21	5.25%
Household Monthly Income	<PKR 50,000	87	21.75%
	PKR 50,000–PKR 250,000	137	34.25%
	PKR 250,000–PKR 500,000	109	27.25%
	>PKR 500,000	67	16.75%
Social Media Usage (per day)	<1 h	63	15.75%
	1–3 h	162	40.50%
	3–5 h	117	29.25%
	>5 h	58	14.50%

**Table 2 behavsci-15-00675-t002:** Reliability and AVE.

Measure	Mean	SD	Cronbach’s Alpha	Composite Reliability	Average Variance Extracted (AVE)
False Self-Presentation	4.85	1.26	0.906	0.906	0.727
Social Comparison	5.13	1.35	0.870	0.872	0.659
Fear of Negative Evaluation	4.51	1.45	0.936	0.937	0.796
Self-Esteem	5.36	1.17	0.825	0.830	0.741
Excessive Social Media Use	4.91	1.52	0.740	0.746	0.563

**Table 3 behavsci-15-00675-t003:** Discriminant validity (HTMT values).

Measure	False Self-Presentation	Social Comparison	Fear of Negative Evaluation	Self-Esteem	Excessive Social Media Use
False Self-Presentation	-				
Social Comparison	0.500	-			
Fear of Negative Evaluation	0.412	0.409	-		
Self-Esteem	0.526	0.647	0.362	-	
Excessive Social Media Use	0.275	0.405	0.379	0.271	-

**Table 4 behavsci-15-00675-t004:** R^2^ values.

Measure	R-Square	R-Square Adjusted
Fear of Negative Evaluation	0.145	0.144
Self-Esteem	0.304	0.303
Excessive Social Media Use	0.154	0.151

**Table 5 behavsci-15-00675-t005:** Moderating role of gender (multi-group analysis).

Relationship	Gender	Path Coefficient	t-Value	*p*-Value	Moderation Effect
False Self-Presentation → Fear of Negative Evaluation	Male	0.370	4.20	0.060	Not Significant
	Female	0.350	3.90	0.070	
Social Comparison → Self-Esteem	Male	0.410	4.50	0.002	Significant
	Female	0.600	6.70	0.000	

## Data Availability

The data supporting the reported results are not publicly available due to ethical and privacy restrictions. However, they can be provided upon reasonable request to the corresponding author, subject to confidentiality agreements.

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
