# Peer review of "The Role of False Self-Presentation and Social Comparison in Excessive Social Media Use"

_behavsci, 2025, doi:10.3390/bs15050675_

Round 1

Reviewer 1 Report

Comments and Suggestions for Authors

The article “The Role of False Self-Presentation and Social Comparison in Excessive Social Media Use” deals with a relevant and current topic. It deals with the relationships between false self-presentation, fear of negative evaluation and excessive social media use as well as between social comparison, self-esteem and excessive social media use, taking gender effects into account. The manuscript utilizes self-discrepancy theory, social comparison theory, and gender schema theory. Overall, the manuscript is well structured and the study was well conducted and provides interesting insights. Nevertheless, I have some questions and suggestions for improvement and clarity. I will list my comments chronologically along the manuscript.

Firstly, although the summary is very long, only two of the three theories used in this paper are mentioned. I would also mention gender schema theory here. In addition, the suggestions for interventions at the end are valuable, but do not quite match the implications at the end of the manuscript. I would make this more coherent and focus on one or two main themes so as not to introduce too many topics. I will come back to the suggestions for action when I comment on the practical implications section.

A general concern, which to some extent also refers to the abstract, is the prominent role of mental health. I agree that social media use and all the variables tested in the model can have an impact on mental health. The authors also show this in terms of some of the literature presented in the introduction and literature review. However, when I first read the summary, I was expecting mental health to be measured here as well. Since this is not the case and we can only speculate that the variables measured may have an impact on mental health, I would go through the entire document carefully and mitigate the role of mental health. With this in mind, can we really say that fear of negative evaluation or overuse of social media equates to poorer mental health? I would be cautious about the impact on mental health, especially for relationships, which were not measured in the current study. It seems fair to speculate in the discussion, but then it needs to be very clear which statements are based on previous research and which are based on the actual data from the current study.

Another concern relates to the “excessive use of social media”. What exactly is excessive? Is it the quantity or the quality of social media use? Is there a definition? I would offer some thoughts on this.

The authors already present relevant findings in the introduction, so that the reader gets a good overview of important contexts. However, it could be made clearer at the end of the introduction where exactly the research gap lies and what the authors contribute in addition to the existing literature.

I don't think it's easy to offer/suggest interventions that are specifically targeted at a particular gender (last sentence of the introduction). This would be very difficult in terms of psychological versus social versus biological gender and gender debates about non-binary genders and LGBTQ-community. I will come back to this later.

On page 4 (l. 160), an interesting thought is expressed, namely that users not only feel bad when they compare themselves with others on social media, but also when they compare their idealized / false self on social media with their real / offline self (if I have understood this correctly). This would also be interesting for the discussion and a deeper analysis of the results.

In general, but especially in the section on self-presentation, I get the impression that the authors equate a false self-presentation with an idealized self-presentation. Of course, this is often the case (especially when referring to the examples the authors give). However, it might be good to mention that idealized is not always completely false, but sometimes “just a little better”. I know the lines are blurred here, but a sentence reflecting on this issue could add depth to the manuscript.

I would recommend writing an introductory sentence before the hypotheses are stated (e.g., “Based on the reported results, we derive the following hypotheses”). I would also indicate the expected direction of the hypotheses (e.g., negatively/positively influenced or increased/decreased).

In the section on social comparison, it seems that all users portray themselves positively on social media. Although positive (/flase) self-presentation is the focus of this study, it could also be mentioned that some people (even strategically) present themselves in a negative light, e.g., to gain support or compassion.

I am a little concerned about the assumptions of the gender schema theory. In my view, the assumptions reflect strong stereotypes. I would recommend reflecting critically on the theory, including in relation to the non-binary gender aspects mentioned and the understanding of psychological gender.

In Figure 1, I would write “false self-representation” instead of “self-representation”. I would also integrate the expected directions (+/-) of the hypotheses. Also, the arrows pointing to gender would have to point in the other direction to clarify the moderating role of gender.

Regarding the method: I was wondering if there were no outliers or participants who quit the survey early. Was the data not cleaned and were there more than the 400 participants who were eventually included in the analysis? Did the participants receive incentives?

Findings:

As far as I know, the description of the sample (age, gender, etc.) is not part of the results, but a separate section (“sample”). Perhaps it could be moved out of the results. In addition, it would be good to know the overall mean and SD for age.

I was wondering why household income was measured and reported. Was there an exploratory analysis on this?

When reporting the results of the structural model evaluation, the authors already give many interpretations of the results. I'm more used to just giving the results in the results section and doing the interpretation in the discussion. I would look at the journal guidelines and some other papers from the journal to get an idea of the usual approach. I would also refer to the specific hypothesis when presenting the results (as is done in the “moderating role of gender” section).

Figure 2 is not easy to read. I would not take the figure from the statistics program, but create it myself (also taking into account the comments on Figure 1).

Discussion:

In the discussion, I would look more critically at the results in relation to gender, referring not only to gender schema theory but also to other literature on gender effects in social media. I would also explain why the findings suggest that women are more affected by social comparison processes, especially in terms of appearance and lifestyle. (p. 14). This sounds like a stereotype to me.

In the discussion, the first few paragraphs state that the findings on the “significant relationship between false self-presentation and fear of negative evaluation” are the most important finding. Why is this the case? Is this result more important than the other results? This would be particularly relevant in light of the statement “One of the most important contributions of this study is the exploration of gender as a moderating factor” on p. 14. It should be clear what the most significant/relevant factor is and stick to this and explain why this is the case.

I would recommend to discuss the findings in chronological order (instead of starting with H2, then reporting H1).

In the discussion, it is not always clear which results and explanations can be derived from the study conducted and which are based on earlier studies. For example, p. 12, lines 483-485, where the reader once again gets the impression that mental health/anxiety and emotions were measured in this study. I would make it very clear what was measured here and to what extent this may be an indication of other variables or previous findings. Overall, I would recommend using more references in the discussion. The authors have provided good literature in the introduction and in the literature review. I would refer back to this and clearly highlight the sources.

Implications:

The authors mention many implications and possible actions to create more awareness and support measures. However, as mentioned in the summary, in my opinion there are too many ideas that are not explained in detail. For example, the term “body positivity” appears for the first time. I also think it is difficult to develop gender-specific measures and also to introduce functions that allow users to filter content that promotes negative emotions. How are users or the platform supposed to recognize which content is too negative? I think this is a very ambitious (but not bad) idea. But it would need more explanation. I would recommend focusing on 1-2 key implications and explaining them in more detail,

Minor:

I would add the variables from the model to the keywords. Also, I would stress that it is about false self-presentation and less about self-presentation in general. Perhaps it would also be good to include a keyword referring to the statistical method.

Some references are not in the correct format (e.g., p. 2 “J.(Jason)”, “C. Yang”, or p. 1 “D. Yang”)

In the section “instruments” the authors write “to answer the research questions…”. However, they stated hypotheses instead of research questions.

Statistical abbreviations (e.g., n, N, p) are not italic, but according to APA it must be italic. I would check the journals guidelines on this.

In table 2, I would equalize the number of decimal places (e.g., for all M and SD 2 digits)

I would always insert the statistical abbreviation, e.g., for “0.213” in “(0.213, p = 0.000)” (p.10)

Overall, I think this is a very important work. Although I made many suggestions, I see much merit in this paper. I wish the authors the best of luck!

Reviewer 2 Report

Comments and Suggestions for Authors

Summary: The authors investigated the psychological factors that may affect excessive social media use, focusing on false self-presentation and social comparison through the lens of Self-Discrepancy Theory and Social Comparison Theory. The study surveyed 400 adult social media users in Pakistan and analyzed the data using Partial Least Squares Structural Equation Modelling. Findings indicated a nuanced landscape of psychological factors that may impact social media behavior that are not limited to habitual use.

Hypotheses:

H1: False self-presentation affects excessive social media use. (not substantiated)

H2: False self-presentation affects fear of negative evaluation. (substantiated - positive)

H3: Fear of negative evaluation affects excessive social media use. (substantiated – positive)

H4: Social comparison affects excessive social media use. (substantiated)

H5: Social comparison affects self-esteem.  (strongly substantiated)

H6: Self-esteem affects excessive social media use. (not substantiated)

H7: Gender moderates the relationship between false self-presentation and fear of negative evaluation, such that the relationship will be stronger for women than for men.

H8: Gender moderates the relationship between social comparison and self-esteem, such that the relationship will be stronger for women than for men. (for both H7/8 (Gender played a moderating role in the relationship between social comparison and self-esteem, with the link being stronger for women than for men. This suggests that women's self-esteem is more susceptible to the effects of social comparison. Though, gender was not a moderator in the relationship between false self-presentation and fear of negative evaluation, indicating similar psychological effects across genders.)

Evaluation:

The most glaring issue here concerns the statements of hypotheses 1-6. The hypotheses could be more clearly worded to address specifics. For example, you state, “H1: False self-presentation affects excessive social media use”. What is the proposed effect? You address it directly in the results and discussion, but it needs to be clearer for the first six hypotheses.

The study explicitly applies Self-Discrepancy Theory and Social Comparison Theory to explain the psychological factors influencing excessive social media use, providing a strong theoretical foundation for the research questions and hypotheses.

Sampling and survey method were both adequate and appropriate for examining relationships between psychological constructs and behavior in a relatively large sample, though there are some concerns as to the self-reported nature of the data in the study. The study employed established instruments and appropriately tested for study reliability. The use of Partial Least Squares Structural Equation Modelling is suitable for testing complex models with multiple constructs and is robust with smaller sample sizes. The analysis included evaluating the measurement model and testing the structural model to assess hypothesized relationships. The use of multi-group analysis to assess moderation is appropriate.

The discussion section connects the findings back to the theoretical framework, explaining the results in the context of Self-Discrepancy Theory, Social Comparison Theory, and Gender Schema Theory. The authors adequately address limitations of the study and provide future directions.

Overall Assessment: The study appears to be well-designed and executed, employing appropriate methodologies and analyses to address its research questions. The use of established theories, validated instruments, and rigorous statistical techniques strengthens the findings. The authors' acknowledgement of the study's limitations and suggestions for future research further demonstrate a thorough and critical approach.

While the acknowledged limitations, particularly the self-report data, are important to consider when interpreting the results and generalizing the findings, they do not detract significantly from the overall quality of the study within its stated aims and scope.

Author Response

Thank you for your feedback. We have made the changes in the hypotheses statement. Looking forward!

Round 2

Reviewer 1 Report

Comments and Suggestions for Authors

The manuscript “The Role of False Self-Presentation and Social Comparison in Excessive Social Media Use” has improved after this revision. I appreciate that the authors considered most of my suggestions and recommendations. However, in my opinion, there are still some aspects that could be improved.

I like the revised version of the abstract. One small suggestion would be to make the last sentence more specific: “Interventions should prioritise digital literacy programs...”. I would specify what is meant by interventions/why interventions are necessary. For example: “Interventions to mitigate/reduce/prevent negative effects of false self-presentation/excessive social media use due to... should prioritize digital literacy programs”.

I also appreciate the additional sentence on p. 3 (ll. 103-106). However, I have the feeling that the sentence here doesn't fit well into the flow of the reading. Perhaps it would fit better after the first sentence on p. 3?

I think that the authors addressed my comment from the previous review (“On page 4 (l. 160), an interesting thought is expressed, namely that users not only feel bad when they compare themselves with others on social media, but also when they compare their idealized / false self on social media with their real / offline self (if I have understood this correctly). This would also be interesting for the discussion and a deeper analysis of the results.”) on p. 5 in ll. 187-193 (although I don't know for sure, since the response letter was a table without my original comments and no line or page numbers, unless I missed something). I appreciate that the authors addressed this issue. However, perhaps this would be better placed elsewhere, e.g., in l. 174? There it could be combined with the points the authors originally made on this topic.

I like the new paragraph on false vs. idealized self-presentation on p. 6 (ll. 245-251). However, again, I feel that it should be provided earlier, when introducing idealized and false self-presentation on p. 5 after line 204.

Although the authors have considered my recommendation to look more critically at the assumptions of gender schema theory in the limitations, I am still somewhat concerned about its use in the theory and discussion sections. I would offer some critical thoughts here as well. Not in the sense that the theory is not appropriate at all, of course (because that would call into question its use in the present work), but in the sense that if it really is the case that self-presentation (concerns) is so stereotypically distributed among men and women, we definitely need more awareness about the different roles that men and women (and all other genders) can take on.

Figure 1 was adapted; however, the direction of the moderating arrows is still not correct. It must point from gender to the arrow between false self-presentation and fear of negative evaluation as well as from social comparison to self-esteem.

It is great to see more information on the sample and data exclusion now. However, I wouldn't just attach it to the end, but integrate it beforehand when stating that there were 400 valid cases (p. 8, l. 343).

I cannot find the section about the samples’ descriptive values anymore (age, gender, education, etc.). Was it excluded? And if so, why? I know that I suggested to move it to another place but it should not be deleted.

The revised paragraphs of 4.2. are clearer now. However, I would include the hypotheses (e.g., “H1) after the respective results for better orientation.

The discussion has also improved. However, I still have some suggestions, very closely related to my comments in the previous review.

Some explanations for the results of this study are very similar to the results of previous studies. Of course, this is not a bad thing per se, but it must be clearly pointed out. For instance, the explanation “This discrepancy triggers negative emotions, particularly anxiety and fear of criticism…” (p. 12, ll. 495-501) was already introduced in the theoretical background and cited with work by Ali et al., 2021, 2023; Bastrygina et al., 2024; Stsiampkouskaya et al., 2021 and Casale et al., 2020; Gill, 2021; Steinert et al., 2025:

“This drive to manage one's online image creates pressure to conform to expected standards, leading to a heightened fear of negative evaluation (Ali et al., 2021, 2023; Bastrygina et al., 2024; Stsiampkouskaya et al., 2021).” (p. 2)

“In addition to its impact on social media usage, false self-presentation also intensifies the fear of negative evaluation. The more effort individuals put into curating an idealized or false version of themselves, the more anxious they become about being judged or criticized for failing to meet those expectations (Casale et al., 2020; Gill, 2021; Steinert et al., 2025).” (p. 5).

Therefore, I would suggest to write something like “in line with previous findings by…”. The same is true for the discussion of H3, H5 (the “common belief” should be more specified and cited), and H6 (the explanations on balanced self-presentation by users with high self-esteem and validation seeking for users with low self-esteem was reported in the theory; see p. 6, ll. 281-287).

There are indeed examples where this was done well, for instance: “This aligns directly with Higgins' Self-Discrepancy Theory (1987), …”(p. 12), or “This aligns with prior work by Meier and Johnson (2022),…” (p. 13).

The implications are now clearer. However, the “digital literacy” comes somewhat suddenly, although it was stated as an implication in the abstract. Perhaps a sentence earlier, stating that the current results shed light on why users use social media excessively and that this can have negative consequences. To avoid these consequences, literacy comes into play.

One last general thought (sorry I didn't see it in the first round of review): Perhaps sections 2.1. and 2.2. should be swapped? After all, self-presentation is the overarching theme and section 2.1. is already about self-presentation. But I leave this decision to the authors and editor (as with all final decisions, of course). 

Minor:

“self – presentation” (p. 12, l. 495) must be without spaces.

The sentence “This is contrary to the common belief that comparing oneself to others leads to feelings of inadequacy.” is redundant to the sentence before it (“Contrary to the widely held belief that comparing ourselves to others often leads to feelings of inadequacy”).

Overall, I continue to believe that this manuscript can provide valuable insight into the connections between false self-presentation, social comparison and excessive social media use. Nevertheless, I think the manuscript would still benefit from a little more clarity and precision.

I wish the authors the best of luck!

Author Response

Response (Round 2)

Reviewer Comment

Change Made

Location of Change

Make the final sentence of the abstract more specific by clarifying why interventions are needed.

The last sentence of the abstract was revised to explain that interventions are necessary to reduce the emotional distress caused by false self-presentation and social comparison, especially in appearance-driven contexts.

Abstract, Page 1

The sentence on p. 3 (ll. 103–106) does not fit well; suggest moving it after the first sentence.

The sentence defining excessive social media use was moved directly after the opening sentence in the paragraph to improve flow.

Page 3, Line 104

Sentence about comparing one’s false and real self could be better placed around l. 174.

Moved the sentence about internal comparison between false and real self to align with related theoretical discussion under Self-Discrepancy Theory.

Page 5, Line 174

Paragraph on false vs. idealised self-presentation (p. 6, ll. 245–251) should be introduced earlier.

The paragraph explaining false vs. idealised self-presentation was relocated to immediately follow the introduction of the two terms.

Page 5, Line 204

Critically reflect on Gender Schema Theory in theory and discussion sections.

Added a paragraph discussing the binary limitations of Gender Schema Theory and the importance of recognising non-binary and fluid gender roles.

Pages 7 & 21 (Sections 2.4 and 7)

Moderating arrows in Figure 1 are incorrect — they must point to the relationships, not the variables.

Revised Figure 1 to depict moderation arrows pointing to the paths between variables (FSP→FNE and SC→SE) using dashed arrows.

Figure 1, Page 8

Integrate sample exclusion details where final sample size is mentioned.

Inserted explanation about data exclusion immediately after the statement of 400 valid responses.

Page 9, Line 343

Descriptive sample statistics (age, gender, education, etc.) are missing.

Reinserted demographic profile in a new subsection '3.2 Demographic Information' including gender, age, education, income, and social media usage.

Section 3.2, Pages 11–12

Label hypotheses (e.g., H1) next to each result in Section 4.2.

Added hypothesis labels (H1–H6) directly after corresponding result statements for clarity.

Section 4.2, Page 13

Clearly distinguish when interpretations in the discussion align with previous findings.

Inserted phrases like 'as previously observed' or 'in line with prior findings' with relevant citations in H2, H3, H5, and H6 discussions.

Pages 20–21

Add a transition sentence before introducing digital literacy in the Implications section.

Added a sentence connecting the study’s psychological findings to the rationale for digital literacy interventions.

Page 22, Section 6

Consider swapping Sections 2.1 and 2.2 to lead with self-presentation.

Retained current order (Self-Discrepancy Theory before False Self-Presentation) to preserve theoretical flow supporting hypotheses H1–H3.

Section 2, Pages 3–6

Correct 'self – presentation' on p. 12, l. 495 to remove spacing.

Corrected formatting to 'self-presentation' with proper hyphenation.

Page 20, Line 495

Remove redundant sentence repeating 'common belief' about comparison and inadequacy.

Deleted the repeated sentence to avoid redundancy.

Page 21, H5 Discussion

Round 3

Reviewer 1 Report

Comments and Suggestions for Authors

The manuscript “The Role of False Self-Presentation and Social Comparison in Excessive Social Media Use” has improved again after this round of revisions. I appreciate that the authors have taken most of my suggestions and recommendations into account.

There are only a few minor aspects that I would like to comment on.

In my last review, I recommended moving the paragraph on the definition of excessive social media use (formerly p. 3; ll. 103-106) after the (former) first sentence on p. 3 (which would have been after “Social comparison is another significant factor contributing to excessive social media use”). The authors have moved the sentence before the recommended position. I would move the sentence “in this study, excessive social media use…” (current document: p. 3, ll. 97-100) after the sentence “Social comparison is another…” (current document: p. 3, ll. 101-102). This is actually only a minor detail, but I think that the new shift will have an enormous benefit for the reading flow and comprehensibility.

In addition, to introduce the topic even better, I would use the word “excessive” even at an earlier place, e.g., in brackets before “social media use” on p. 2 in line 50: “In particular, (excessive) social media use has been linked to...”

After reading the sentence “Selfies presentation, for instance is a very prominent and popular feature…” (pp. 2-3, ll. 91-94) again, I’m a little confused. I don’t think I understand the essence of this sentence. Why does it say “in this study”? Were selfies a central aspect of the measures? I'm sorry I didn't mention this before, but I think a little rewording should fix this problem. In line with this, it would be helpful to include example items in the measures section.

I appreciate that the authors follow my suggestions for moving paragraphs. However, sometimes the transitions between the newly inserted paragraphs also need a little adjustment. For example, on p. 4, l. 181: “this curation” is now separated from the earlier curation-part (now in l. 170, same page). Therefore, I would recommend adjusting the introduction of the sentence a little to ensure a good reading flow, e.g.: “the curation of self-presentation can exacerbate the gap between the actual self and the ideal self...”

I really like the new paragraph on the Gender Schema Theory on p. 7!

I'm also pleased to see the descriptive values again. However, I still wonder why income was measured and reported. Perhaps a sentence would be helpful here as an explanation (see my first review). In addition, as stated in my first review, it would be good to know the overall mean and SD for age (in addition to age range percentages).

Overall, I applaud the efforts that the authors have made since the first submission. I would also encourage them not to “just do what the reviewer says”, but - if necessary - to reflect critically on the claims and/or explain why some changes should not be made/why some information is given or not given. I hope this doesn't sound patronizing, but is just meant as friendly advice.

I think this is a great work!

I wish the authors the best of luck!

Minor:

All theories should be consistently written with capital letters in the beginning. On p. 4 (l. 161), for instance, “self-discrepancy theory” is written in lower case. Also, in the description of Figure 1, “Self presentation theory” and “gender theory” are not written with capital letters in the beginning.

Author Response

We sincerely thank the reviewer for their continued engagement and thoughtful feedback, which have substantially strengthened this manuscript. We have carefully addressed every remaining concern and have revised the document accordingly to ensure improved clarity, coherence, and scholarly consistency.

First, we have repositioned the sentence defining excessive social media use to follow the recommended placement—immediately after explaining social comparison (pg. 3, line 106). Additionally, we have introduced the word “excessive” earlier in the manuscript (p. 2, l. 50) to sharpen the focus from the outset, as suggested. To clarify a previously confusing sentence about selfies, we removed the misleading phrase “in this study” and revised the paragraph to explain the relevance of selfies in relation to self-presentation literature (pg. 2, line 91). We have also included example items for each scale used in the study within the Instruments section to increase transparency and specificity.

We carefully reviewed the entire manuscript to ensure all theories are capitalised consistently. Transitional clarity has also been improved. Finally, in response to the comment regarding demographic variables, we have added a rationale for reporting household income and provided the overall mean and standard deviation for age (M = 28.4, SD = 8.3) to offer a more complete profile of the sample.

We are grateful for the reviewer’s encouraging remarks and critical insights throughout this process.